# Protocol for a multicentre randomised controlled trial of STeroid Administration Routes For Idiopathic Sudden sensorineural Hearing loss: The STARFISH trial

Matthew E. Smith[1,2]*, Rachel Knappett[2], Deborah Vickers[1], David White[3], Chris J. Schramm[4], Samir Mehta[5], Yongzhong Sun[5], Ben Watkins[5], Marie Chadburn[5], Hugh Jarrett[5], Karen James[5], Elizabeth Brettell[5], Tracy E. Roberts[5], Manohar L. Bance[1,2], INTEGRATE (the UK ENT Trainee Research Network)[¶], James R. Tysome[1,2]

1 University of Cambridge, Cambridge, United Kingdom, 2 Cambridge University Hospitals NHS Foundation Trust, Cambridge, United Kingdom, 3 Patient Advocate, United Kingdom, 4 Independent General Practitioner, NHS General Practice, United Kingdom, 5 Birmingham Clinical Trials Unit, University of Birmingham, Birmingham, United Kingdom

¶ The UK ENT Trainee Research Network, www.entintegrate.co.uk (GROUP AUTHOR).
* mes39@cam.ac.uk

**Data Availability Statement:** No datasets were generated or analysed during the current study. All

## Abstract

Idiopathic sudden sensorineural hearing loss (ISSNHL) is the rapid onset of reduced hearing due to loss of function of the inner ear or hearing nerve of unknown aetiology. Evidence supports improved hearing recovery with early steroid treatment, via oral, intravenous, intratympanic or a combination of routes. The STARFISH trial aims to identify the most clinically and cost-effective route of administration of steroids as first-line treatment for ISSNHL. STARFISH is a pragmatic, multicentre, assessor-blinded, three-arm intervention, superiority randomised controlled trial (1:1:1) with an internal pilot (ISRCTN10535105, IRAS 1004878). 525 participants with ISSNHL will be recruited from approximately 75 UK Ear, Nose and Throat units. STARFISH will recruit adults with sensorineural hearing loss averaging 30dBHL or greater across three contiguous frequencies (confirmed via pure tone audiogram), with onset over a ≤3-day period, within four weeks of randomisation. Participants will be randomised to 1) oral prednisolone 1mg/Kg/day up to 60mg/day for 7 days; 2) intratympanic dexamethasone: three intratympanic injections 3.3mg/ml or 3.8mg/ml spaced 7±2 days apart; or 3) combined oral and intratympanic steroids. The primary outcome will be absolute improvement in pure tone audiogram average at 12-weeks following randomisation (0.5, 1.0, 2.0 and 4.0kHz). Secondary outcomes at 6 and 12 weeks will include: Speech, Spatial and Qualities of hearing scale, high frequency pure tone average thresholds (4.0, 6.0 and 8.0kHz), Arthur Boothroyd speech test, Vestibular Rehabilitation Benefit Questionnaire, Tinnitus Functional Index, adverse events and optional weekly online speech and pure tone hearing tests. A health economic assessment will be performed, and presented in terms of incremental cost effectiveness ratios, and cost per quality-adjusted life-year. Primary analyses will be by intention-to-treat. Oral prednisolone will be the reference. For the primary outcome, the difference between group means and 97.5% confidence intervals at

relevant data from this study will be made available upon study completion.

**Funding:** This work was funded by a grant from UK NIHR Health Technology Assessment, HTA Reference Number: NIHR131528. The grant was awarded to JRT and MES as joint lead applicants, with co-applicants CS, DV, DW, HJ, SM, RK, MB and TR: https://fundingawards.nihr.ac.uk/award/NIHR131528. The funder provided feedback on study design at an early stage, but has not had a role in protocol development or publication. The primary Sponsor for the study is the University of Birmingham, Edgbaston, Birmingham, B15 2TT. This research was supported by the NIHR Cambridge Biomedical Research Centre.

**Competing interests:** The authors have declared that no competing interests exist.

each time-point will be estimated via a repeated measures mixed-effects linear regression model.

## Introduction

Sudden sensorineural hearing loss (SSNHL) is the rapid onset of reduced hearing due to loss of function of the inner ear or hearing nerve. The cause is found in only 10–15% of participants [1, 2], and in most cases the aetiology is unknown or "idiopathic". Idiopathic sudden sensorineural hearing loss (ISSNHL) is usually unilateral, has an incidence of 5–20 per 100,000 and can result in permanent and complete hearing loss, although spontaneous recovery is seen in 32–65% of cases [3]. ISSNHL can have a profound impact on participants and their quality of life.

Evidence supports improved hearing recovery with early steroid treatment, via oral, intravenous or intratympanic routes [1, 3], with a combination of oral and intratympanic steroids possibly leading to superior hearing recovery [4]. Identification of the most clinically and cost-effective route of administration of steroids as first-line treatment for ISSNHL is a high priority research recommendation from the UK National Institute for Health and Care Excellence (NICE) guidelines for assessment and management of hearing loss in adults [2].

The best route of steroid delivery for hearing recovery in ISSNHL is unknown. Optimising treatment for ISSNHL is important for both participants and health resources as the condition is associated with reduced quality of life [5]. Patient and Public Involvement (PPI) work by the trial team also identified a need for better diagnosis and access to specialist care if the management and hearing outcomes of individuals with ISSNHL are to be improved.

STARFISH is a three arm, superiority randomised, controlled trial that aims to identify the most clinically and cost-effective route of administration of steroids as first-line treatment for ISSNHL. Procedurally, the trial is designed to be in line with standard UK clinical practice as much as possible, to minimise burden on participants. The trial has been designed to establish the effectiveness of the most commonly used interventions for ISSNHL in the UK, using a broad range of outcomes that are important to patients, clinicians and policymakers, measuring functional hearing, associated symptoms, intervention adverse effects and patient quality of life.

## Materials and methods

The STARFISH trial can be found in the ISRCTN registry (ISRCTN10535105, https://doi.org/10.1186/ISRCTN10535105 ). A full version of the protocol is available at https://www.birmingham.ac.uk/starfish and https://entintegrate.co.uk/starfish, along with patient and clinician videos and updated details on trial recruitment. The protocol is also available as supporting information [S1 Protocol]. The SPIRIT checklist used for protocol reporting can be accessed [S1 Checklist].

The STARFISH trial was granted favourable opinion by London - Harrow Research Ethics Committee (REC reference 22/LO/0532, trial IRAS ID 1004878).

### Aim of the study

The aim is to evaluate the clinical and cost effectiveness of oral, intratympanic or combined oral and intratympanic steroids as the first line of treatment for ISSNHL.

## Design and setting of the study

STARFISH is a pragmatic, multicentre, assessor-blinded, parallel, three-arm intervention, superiority, randomised controlled trial (1:1:1) with an internal pilot. Participants treated within the United Kingdom National Health Service will be recruited from approximately 75 secondary or tertiary care Ear, Nose and Throat units. A full list of sites can be found in the ISRCTN registry. A SPIRIT schedule of enrolment, interventions, and assessments is presented as Fig 1, and a flow diagram of the study can be found in Fig 2.

## Inclusion and exclusion criteria

### Inclusion criteria.

- Adults aged 18 years or over

- Diagnosis of new-onset ISSNHL: a new increase in sensorineural thresholds of 30 decibels (dBHL) or greater with onset over a period of 3 days or less according to the patient's history, and affecting each of 3 contiguous pure-tone frequencies (out of 0.5, 1.0, 2.0, 4.0 kilohertz (kHz)) confirmed with a pure tone audiogram. Where audiometry is not available prior to the ISSNHL and there is a history is of equal hearing in both ears prior to the sudden loss, hearing loss will be defined in relation to the opposite ear's thresholds. Where audiometry is not available prior to the ISSNHL and there is a history of different hearing in both ears prior to the sudden loss, then the candidate can only be included if the ISSNHL occurred in the better hearing ear and the measured thresholds are at least 30dB below the contralateral ear at 3 contiguous pure-tone frequencies (out of 0.5, 1.0, 2.0, 4.0 kilohertz (kHz)) confirmed with a pure tone audiogram.

- Onset of hearing loss within four weeks prior to randomisation

- English spoken as a first or second language

### Exclusion criteria.

- Identified cause for hearing loss (not idiopathic)

- Bilateral ISSNHL

- Received prior steroid treatment for the same episode of ISSNHL

- Medical contraindication to high dose systemic steroids

- Previous history of psychosis

- On oral steroid therapy for another condition

- Known adrenocortical insufficiency other than exogenous corticosteroid therapy

- Hypersensitivity to the active substance or to any of the excipients

- Systemic infection unless specific anti-infective therapy is employed

- Ocular herpes simplex

- Ipsilateral acute or chronic active middle ear disease (including acute otitis media, chronic suppurative otitis media and cholesteatoma, excluding dry perforation)

- Does not have the capacity to provide written informed consent

| TIMEPOINT | STUDY PERIOD | | | | | | |
|---|---|---|---|---|---|---|---|
| | Enrolment | Allocation | Post-allocation | | | | |
| | | | | From 1st Injection | | From allocation | |
| | *0* | *0* | *0±3 days* | *7±2 days* | *14±2 days* | *42±7 days* | *84±7 days* |
| **ENROLMENT:** | | | | | | | |
| **Eligibility screen** | All | | | | | | |
| **Informed consent** | All | | | | | | |
| **Relevant medical history recorded** | All | | | | | | |
| **Concomitant medication check** | All | | | | | | |
| **Demographic information recorded** | All | | | | | | |
| **Allocation** | | All | | | | | |
| **INTERVENTIONS:** | | | | | | | |
| **Oral steroid course start** | | | 1,3 | | | | |
| **Intratympanic steroid injection** | | | 2,3 | 2,3 | 2,3 | | |
| **ASSESSMENTS:** | | | | | | | |
| **Otoscopy** | | | All | All | All | All | All |
| **Pure tone audiogram (0.5-8.0kHz range)** | | | All | All | All | All | All |
| **AB phoneme speech testing** | | | All** | | | All | All |
| **Online digits-in-noise test#** | | | All# | ◆————————————————◆ | | | |
| **Online pure tone audiogram test#** | | | All# | ◆————————————————◆ | | | |
| **Speech, Spatial and Qualities of hear scale (SSQ)** | | | All | | | All | All |
| **Rehabilitation Benefit Questionnaire (VRBQ)** | | | All | | | All | All |
| **Tinnitus Functional Index (TFI)** | | | All | | | All | All |
| **Health Utilities Index 3 (HUI3)** | | | All | | | All | All |
| **ICECAP-A** | | | All | | | All | All |
| **Resource usage** | | | | | | All | All |
| **Adverse Events monitoring** | | | | | | All | 2,3& |
| **Compliance reporting** | | | | | | All | |

**Fig 1. SPIRIT schedule of enrolment, interventions, and assessments.** 1 = for arm 1 (oral steroid), 2 = for arm 2 (intratympanic injection), 3 = for arm 3 (combined treatment). # Optional weekly testing, for participants who have internet access. * should be performed within three days prior to commencement of treatment. **recommended to be performed on same day as pure tone audiogram or if not practicable within next working day. & Only persistent perforation of the tympanic membrane will be recorded at 12 weeks.

## Characteristics of participants

Consecutive potentially eligible participants will be identified by clinicians at participating sites following referral to the ENT unit from general practitioners, Emergency Departments,

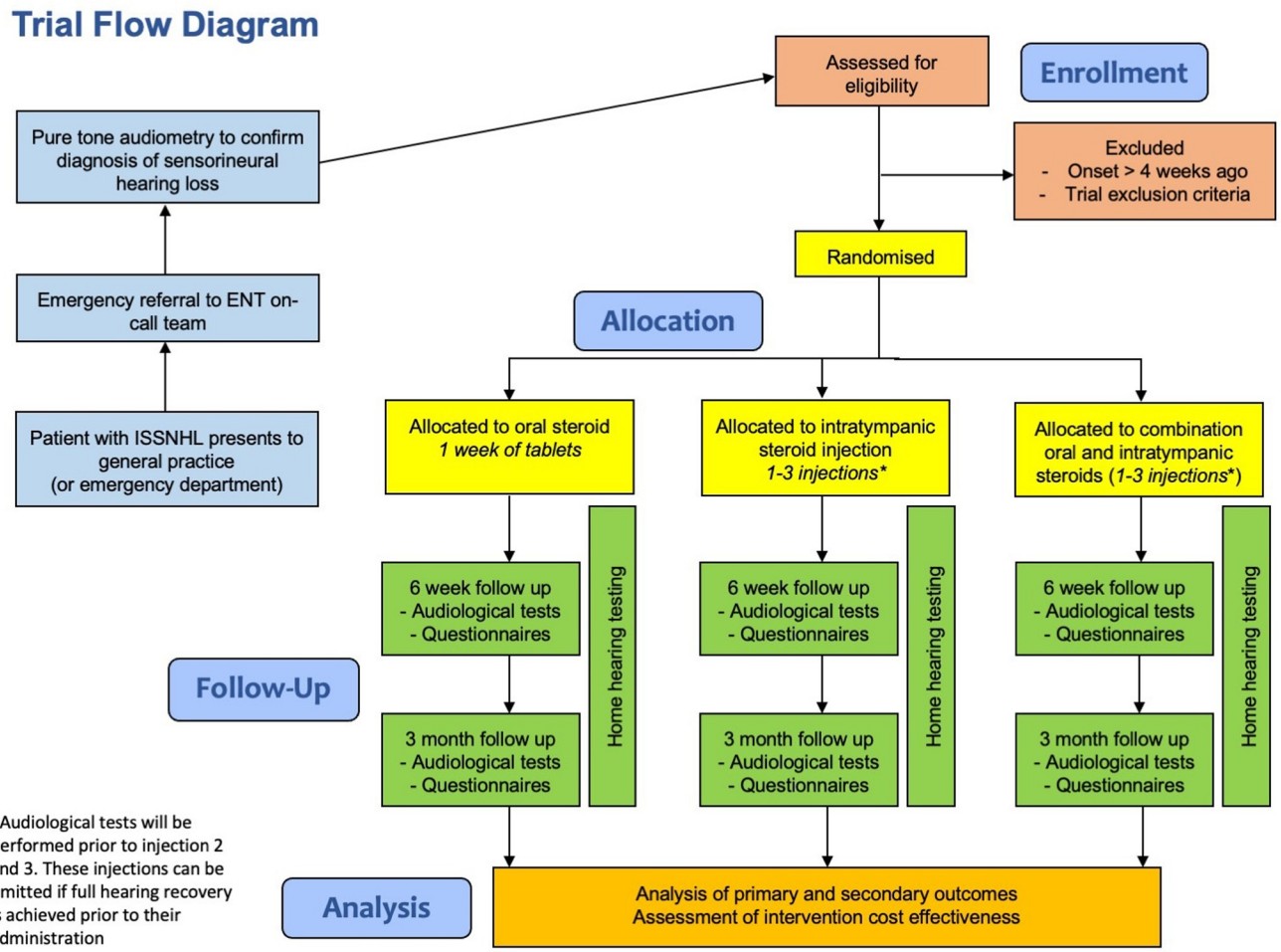

**Fig 2. Flow diagram showing the design of the STARFISH trial and the participant pathway.**

and audiology services. The reported history of the hearing loss and pure tone audiometry will be used as criteria for potential candidates, and a member of the trial team (ENT surgeon, research nurse or audiologist) will then approach the individual about the study. Participants will then be screened using trial eligibility criteria. Details of all participants approached about the trial will be recorded electronically on the Participant Screening Log.

## Randomisation

Using a secure online system, once recruited, participants will be randomised by a member of the trial team using an online randomisation system through the University of Birmingham Clinical Trials Unit at the level of the individual in a 1:1:1 ratio to either oral steroid only, intratympanic steroid only, or combined oral and intratympanic steroid treatment. A minimisation algorithm will be used to ensure balance in the treatment allocation over the following variables:

- Hearing loss severity based on pure tone average (PTA) at 0.5, 1, 2 and 4kHz (mild/moderate less than 70dBHL; severe/profound 70dBHL or greater)

- Time since onset of hearing loss (≤14 days; >14 days)

- New vertigo since onset of hearing loss (reported by participant, yes; no)

- Hospital site

## Interventions

Participants will be randomised to one of three groups:

Group 1) Oral steroid (prednisolone as tablets) 1mg/kg body weight/day up to 60mg/day for 7 days.

Group 2) Intratympanic steroid (dexamethasone as dexamethasone phosphate) three intra-tympanic injections 3.3mg/ml or 3.8mg/ml spaced 7±2 days apart (see below for technique).

Group 3) Combined oral (prednisolone) and intratympanic (dexamethasone) steroid, as described above, with the first intratympanic injection occurring ±4 days of starting oral steroids.

Immediately prior to the second and third intratympanic injections, a pure tone audiogram will be performed. If full recovery has been achieved (return to within 10dBHL of the unaffected ear at 0.5, 1.0, 2.0, 4.0kHz, if contralateral ear has no participant-reported pre-existing loss), the participant and clinician may jointly decide to omit further intratympanic injections.

**Technique for intratympanic injection.** Dexamethasone 3.3mg/ml or 3.8mg/ml will be administered, depending on local policy. The procedure will be performed using a microscope or endoscope to visualise the tympanic membrane, with the participant recumbent at an angle of <20 degrees. Any locally adopted method of local anaesthesia will be used. A 22–25 gauge spinal needle will be passed through the tympanic membrane anterior-inferiorly, and the dexamethasone slowly injected until the fluid level reaches the level of the needle. Following the injection, the study participant will maintain their head turned away from the injected ear, recumbent for 30 minutes to maximise uptake of steroid through the round window. Participants will be provided with information leaflets specific to the intervention(s) they received, detailing precautions following intervention, possible side effects, and where to seek further information if needed [S5 and S6 Files].

Written material and a training video will be issued to investigators to standardise the injection technique, and face-to-face training will be provided where requested by the site.

## Outcomes

The primary outcome will be the absolute improvement in pure tone audiogram average at 12 weeks following randomisation (calculated at 0.5, 1.0, 2.0, 4.0kHz), conducted by an audiologist blinded to the treatment allocation.

Secondary outcomes will be measured at 6 and 12 weeks following randomisation.

**Functional hearing.**

- Hearing related to speech will be assessed using The Speech, Spatial and Qualities of hearing scale (SSQ). SSQ-12 is a validated 12-item questionnaire [6], known to provide a good representation of the functional relationship with speech in everyday life and has been used to assess the disability of unilateral hearing loss seen in ISSNHL [6, 7], providing disability scores associated with different aspects of hearing.

- The absolute improvement in pure tone audiogram average at 6 weeks following randomisation (calculated at 0.5, 1.0, 2.0, 4.0kHz). Conducted by an audiologist blinded to the treatment allocation.

- Hearing thresholds measured by pure tone audiogram average following treatment initiation (calculated at 0.5, 1.0, 2.0, 4.0kHz).

- High frequency hearing thresholds measured by the absolute improvement in pure tone audiogram average across 4.0, 6.0 and 8.0 kHz.

- Recovery of speech perception measured using Arthur Boothroyd (AB) word lists scored by phoneme [8]. This speech testing will be carried out by an audiologist blinded to the treatment allocation. The AB word lists test is the most widely used speech audiometry in the UK and is in routine use as an assessment tool for single sided hearing loss. The assessment will start at 30dB above the PTA average and intensity will be increased until the maximum score is reached.

- Extent of hearing recovery classification (complete/partial/none) based on pure tone and speech audiometry [1] (Table 1).

- Time to hearing recovery/rate of recovery determined using online digits-in-noise and pure tone hearing tests accessed via the trial website (optional and recommended weekly where done). For validation, online test data will be compared to in-hospital audiometry data.

**Associated symptoms.**

- Dizziness will be assessed using the Vestibular Rehabilitation Benefit Questionnaire (VRBQ). This is a 20-item participant completed tool that provides a measure of the anxiety associated with dizziness [9, 10], identified as an important outcome by our patient group.

- Tinnitus will be assessed using the Tinnitus Functional Index (TFI). This is a 25-item participant completed questionnaire that is sensitive to changes in tinnitus severity and impact [11] and is recommended by NICE for this purpose [12].

**Adverse events.**

- Adverse events (AEs) relevant to the interventions will be recorded.

Table 1. Classification of hearing recovery.

| Degree of hearing recovery | American Academy of Otolaryngology–Head & Neck Surgery criteria / Functional change | Outcome criteria |
|---|---|---|
| Unchanged | No recovery | • <10dB change in the pure tone audiogram |
| Partial Recovery Unaidable | Recovery to less-than-serviceable levels indicates an ear unlikely to benefit from traditional amplification. Can use specialist hearing aid technology but unable to restore spatial hearing. | • ≥10dB improvement in pure tone audiogram • AB phoneme maximum score <50% |
| Partial Recovery Aidable | Recovery to a serviceable level typically indicates that after recovery, the ear would be a candidate for traditional hearing amplification | • ≥10dB improvement in pure tone audiogram • AB phoneme maximum score 50% or more |
| Full Recovery | Complete recovery requires return to within 10dBHL of the unaffected ear and | • Return to normal hearing (20dB or lower) or to within 10dB of the contralateral ear |

American Academy of Otolaryngology–Head & Neck Surgery hearing recovery criteria [16]. Classification shown with the associated audiometric criteria.

**Health economic assessment.**

- Two tools will be used to assess health economics: the Health Utilities Index 3 (HUI3), a participant-reported assessment of health-related quality of life suited to hearing loss [13], and ICEpop CAPability measure for Adults (ICECAP-A), a participant-reported measure of capability for the adult population [14, 15].

- Resource usage in the 12 weeks following randomisation will be recorded.

The detailed procedure for outcome assessment can be found in supporting material [S1 File].

## Blinding

Participant blinding is not required given the method of primary outcome assessment, which is objective from the perspective of the participant. The audiologist assessing the primary outcome (the pure tone audiogram) will be blinded to the randomised treatment allocation, with prior assessment of ear suitability for testing by another investigator. Where feasible, the pre-injection audiogram will be conducted by an audiologist not recording later outcomes for the trial.

## Trial procedures

The study has been designed to ensure participant contact correlates to standard care in the UK where possible. Table 2 provides a schedule of interventions and patient assessments. Prior to commencing recruitment, each site will undergo a process of initiation. Investigators will be required to attend a virtual initiation meeting, covering aspects of the trial design, protocol procedures, adverse event reporting, collection and reporting of data, and record keeping.

**Baseline assessment and interventions.** Following randomisation, an investigator (ENT surgeon) will complete an online baseline case report form (CRF), collecting data on demographics, medical history, concomitant medication, pure tone and speech audiometry. The questionnaires (SSQ, VRBQ, TFI, HUI3, ICECAP-A) will be issued in a paper booklet for the participant to complete themself. Given the urgent nature of treatment, interventions will usually begin on the day of randomisation, with the prescribing of oral steroids and delivery of the first intratympanic injection. If intervention is delayed, it must occur within four weeks of symptom onset.

**Follow up assessments.** Follow up for the collection of secondary outcome and health economic data will occur at 6 weeks ±7 days and 12 weeks ±7 days after randomisation. At these points online follow up CRFs will be completed by an investigator, including audiometry (audiologist), adverse events, treatment compliance, and resource use data (ENT surgeon), as indicated in Table 2. The participant-completed questionnaires will again be issued in a paper booklet at both time points.

**Home hearing testing.** For participants who have internet access and agree to use online digits-in-noise and pure tone tests, these will be accessible on the STARFISH trial website. Testing on a weekly basis will be encouraged, for 12 weeks from randomisation with weekly reminders by text message to encourage participation. Online test results will automatically be collated centrally.

## Sample size

Sample size has been calculated based on the primary outcome measure of absolute improvement in pure-tone thresholds. A typical gain in hearing threshold in the control group (oral

**Table 2. Schedule of assessments.**

| Assessments | Screening | Baseline | time from first injection | | | time from randomisation | | |
|---|---|---|---|---|---|---|---|---|
| | | | 2nd injection (week 1 ±2 days) | 3rd injection (week 2 ±2 days) | Weekly | Week 6 ±7 days | Week 12 ±7 days |
| Eligibility check | All | | | | | | | |
| Valid informed consent | | All | | | | | | |
| Relevant medical history taken | | All | | | | | | |
| Concomitant medication check | | All | | | | | | |
| Demographic information | | All | | | | | | |
| Randomisation | | All | | | | | | |
| Otoscopy | | All | 2,3 | 2,3 | | | All | All |
| Pure tone audiogram (0.5–8.0kHz range) | | All* | 2,3 | 2,3 | | | All | All |
| AB phoneme speech testing | | All** | | | | | All | All |
| Online digits-in-noise test# | | | | | All# | | | |
| Online pure tone audiogram test# | | | | | All# | | | |
| Speech, Spatial and Qualities of hear scale (SSQ) | | All | | | | | All | All |
| Vestibular Rehabilitation Benefit Questionnaire (VRBQ) | | All | | | | | All | All |
| Tinnitus Functional Index (TFI) | | All | | | | | All | All |
| Health Utilities Index 3 (HUI3) | | All | | | | | All | All |
| ICECAP-A | | All | | | | | All | All |
| Resource usage | | | | | | | All | All |
| Adverse Events monitoring | | | | | | | All | 2,3& |
| Compliance reporting | | | | | | | All | |
| Oral steroid provision | | 1,3 | | | | | | |
| Intratympanic injection | | 2,3 | 2,3 | 2,3 | | | | |

1 = arm 1 (oral steroid), 2 = arm 2 (intratympanic injection), 3 = arm 3 (combined treatment)

#Optional, for participants who have internet access.

* should be performed within three days prior to commencement of treatment

**recommended to be performed on same day as pure tone audiogram or if not practicable within next working day & Only persistent perforation of the tympanic membrane will be recorded at 12 weeks

steroid only) would be 25 dBHL, with standard deviation (SD) of 25 dBHL [3]. As specified in published international criteria for ISSNHL [2], we have defined 10 dBHL as a minimum clinically important difference (MCID) for the sample size. With 90% power and alpha of 0.025 (for the 3-arm trial design with two key group comparisons *[see statistical analysis section for further details on this]*), to observe a MCID of 10 dBHL with SD of 25 dBHL, the trial requires 157 participants per arm, or 471 participants in total. After allowing for a 10% attrition rate this gives a sample size of 175 per arm, or 525 in total for a 3-arm study.

## Safety considerations

The interventions specified in the STARFISH trial are all routinely used in standard care within the UK. A risk assessment has been conducted and concluded that this trial can be categorised as representing no higher risk than standard medical care.

The recording and reporting of adverse events (AEs) will be in accordance with the UK Policy Framework for Health and Social Care Research, the Principles of Good Clinical Practice, as set out in the UK Statutory Instrument (2004/1031; and subsequent amendments), the

requirements of the Health Research Authority and The Medicines for Human Use (Clinical Trials) Regulations 2004 and amendments thereof. The process for AE recording and reporting, and the definitions for different categories of AEs are detailed in supporting material [S2 File].

Personal data will be regarded as strictly confidential and will be handled and stored in accordance with the Data Protection Act 2018 (and subsequent amendments).

## Internal pilot

The success of the nine-month internal pilot study will be based upon the recruitment rate, adherence to the treatment allocation and dropout rate. At nine months, it is anticipated that the trial will have recruited 81 participants from at least 36 centres. For the trial to continue, the pre-defined stop-go criteria must be met, and a traffic light system has been designed to determine continuation to the full trial:

**Green:** Recruitment rate ≥100%, adherence rate ≥90%, and dropout rate <5%. If all three criteria are met, the trial will continue with the protocol unchanged.

**Amber:** Recruitment rate 70–99%, adherence rate 70–89%, or dropout rate 5–10%. If one or more of the criteria are met, the trial will be reviewed to determine the changes (if any) that could be made to improve criteria not at the green level.

**Red:** Recruitment rate <70%, adherence rate <70%, or dropout rate >10%. If one or more of these criteria are met, this will be discussed with the Trial Steering Committee and funder regarding feasibility of the trial continuing.

## Data management plan

Processes will be employed to facilitate the accuracy and completeness of the data included in the final report. Data entry will be completed by the site staff via a bespoke online trial database, with the exception of participant-completed booklets data, which will be transcribed centrally. Automatic range checks and data review will ensure high levels of data quality. Overdue data entry and data queries will be requested from sites. All data records created for the trial will be stored securely and confidentially for at least 25 years. At the time of publication, anonymised trial data will be made available by formal application to a data sharing committee.

## Statistical analyses

The reporting of the outcomes will be based on published guidelines [17]. The objective of the trial is to test the superiority of three interventions, and in all analyses the oral steroid group will be considered as the reference category. All primary analyses (primary and secondary outcomes including safety outcomes) will be by intention-to-treat, i.e. all participants will be analysed in the intervention group to which they were randomised irrespective of adherence to randomised intervention or other protocol deviation. As a secondary (or sensitivity) analysis, a per-protocol analysis will also be carried out for the primary outcome.

For the primary outcome and all other secondary outcomes measures, a Bonferroni correction [18] will be applied to account for the increase in the risk of type I error associated with making two key comparisons (i.e. "intratympanic steroid versus oral steroid" and "combination oral and intratympanic steroid versus oral steroid"). To maintain an overall 5% type I error rate, each comparison will be tested at a significance level of 2.5%. A comparison of intratympanic steroid versus combination oral and intratympanic steroid will also be performed as part of an exploratory analysis for all outcomes.

For all outcome measures, appropriate summary statistics and differences between groups, e.g. mean differences, relative risks, absolute differences will be presented, with two-sided

97.5% confidence intervals. Outcomes will be adjusted for the minimisation variables and any baseline values (where applicable).

For the primary outcome, the difference between group means and associated 97.5% confidence intervals at each time-point will be estimated through the use of a repeated measures mixed-effects linear regression model. All assessment times (baseline and the post-treatment time points) will be included. Parameters allowing for treatment group, time variable, the randomisation minimisation variables except for site and the baseline score will be included as fixed effects and participants nested within site as random effects. Time will be assumed to be a categorical (fixed) variable. To allow for a varying treatment effect over time, a time by treatment interaction parameter will be included in the model. A compound symmetry covariance structure will be assumed. Estimates of differences between groups at the 6 week and 12 week time-points will be taken from the model including this interaction parameter. Results will be presented as adjusted mean difference and 97.5% confidence interval. Subgroup analyses will be limited to the same variables used in the minimisation algorithm except for site and performed on the primary outcome only.

The secondary outcomes consist of continuous, time to event, binary, ordered categorical with more than two categories and count data types. For all secondary outcomes that are continuous, they will be analysed using the same statistical method as described for primary outcome. All analysis will be adjusted for the minimisation variables as fixed effects (except site which will be included as a random effect).

For those secondary outcomes that are binary, data will be summarised as number and percentage of participants in each intervention arm and an adjusted relative risk and 97.5% confidence interval will be estimated from a log-binomial regression model for each comparison. For secondary outcomes that are ordered categorical with more than two categories, data will be summarised as number and percentage of participants in each category by intervention arm. An adjusted odds ratio and 97.5% confidence interval will be estimated from an ordered logistic regression model for each comparison. Any secondary outcomes that are time to event data will be compared between intervention arms using survival analysis methods. Kaplan-Meier survival curves will be constructed for visual presentation of time-to-event comparisons. A Cox proportional hazards model will be fitted to obtain the adjusted hazard ratios and 97.5% confidence intervals. For secondary outcomes that are count data type, they will be analysed using a Poisson regression model (or negative binomial regression if there is evidence of overdispersion) with an offset for the length of time the participant was in the trial being included in the model, to obtain an adjusted incidence rate ratio and 97.5% confidence interval for each comparison. Statistical analysis will be undertaken in Stata version 18 (or higher) and/or SAS software, version 9.4 (or higher).

## Definition of adherence

Adherence to oral steroid will be monitored at the 6 week follow up visit and defined as taking prednisolone every day for 7 days, with treatment initiated within 4 weeks of symptom onset, and not having the intratympanic treatment.

Adherence to intratympanic treatment group will be defined as receiving 3 intratympanic injections of dexamethasone (either 3.3mg/ml or 3.8mg/ml) spaced 7+/-2 days apart, unless the course of injections was reduced to 1 or 2 injections on the basis of pure-tone audiogram-confirmed resolution of hearing loss or evidence of further sensorineural hearing loss, as well as not having received any oral steroid, with treatment initiated within 4 weeks of symptom onset. A reduction in course duration for intratympanic injection for any other reason will not be considered per-protocol.

Adherence to the combined treatment group will require the criteria above to be met for both interventions. Additionally, if oral treatment is initiated within 4 weeks of symptom onset then intratympanic treatment must commence up to four days preceding or following this.

**Handling protocol deviations.** A protocol deviation is defined as a failure to adhere to the protocol such as errors in applying the inclusion/exclusion criteria, the incorrect intervention being given, incorrect data being collected or measured, follow-up visits outside the visit window or missed follow-up visits. We will apply a strict definition of the intention-to-treat principle and will include all participants regardless of deviation from the protocol.

This does not include those participants who have specifically withdrawn consent for the use of their data in the first instance. However, these outcomes will be explored as per other missing responses. Most assessments will be conducted by an audiologist blinded to the treatment allocation and it is anticipated that the assessments will be completed within the scheduled time window. For participants who have internet access and agree to use the online digits-in-noise and pure tone tests, they will be asked to perform the tests on the trial website every week, for 12 weeks, from commencement of treatment. The completion dates of these tests will be checked to ensure they fall at the correct time-point.

**Unblinding.** This is not applicable as this is an assessor-blinded study only.

## Health economic assessment

**Within-trial economic evaluation.** A within-trial based economic evaluation will explore the cost-effectiveness of the three interventions being compared. A preliminary cost consequence analysis will be carried out to establish whether any intervention shows dominance. Dominance will require one route of steroid delivery to be less costly with improved outcomes upon which the analysis is based. The economic evaluation will be based on three different outcomes: cost per case of any improvement in hearing; cost per case for the different increments of hearing improvement and; and cost per quality-adjusted life-year. Health-related quality of life will be assessed primarily using the HUI3, which is considered most appropriate in the field of hearing loss [13, 14]. ICECAP-A will be used to assess other attributes of wellbeing. These utility data will be collected at baseline, 6 and 12 weeks. All other outcomes, improvements in hearing and increments of hearing improvement will be collected and assessed at 12 weeks.

**Data collection for the economic evaluation.** Resource use data relating to the hearing loss, interventions and adverse events will be collected prospectively from an NHS perspective, in order to estimate the overall cost of the alternative interventions. Additional resource use such as that associated with additional appointments or interactions with the health service related to the intervention or AEs will also be collected via participant interview at 12 weeks after randomisation. The primary analysis will adopt the perspective of the health service and personal social services. Information on unit costs will be obtained from key UK national sources, such as the NHS reference costs, the Unit Costs of Health and Social Care, the British National Formulary, and the Office for National Statistics [19].

**Presentation of economic evaluation results.** Initially, the base-case analysis for the within-trial analysis will be framed in terms of cost consequences, reporting data in a disaggregated manner on the incremental cost and the important consequences as assessed in the trial. The outcomes for the economic evaluation (Health Utilities Index 3 and ICECAP-A) will be presented in terms of Incremental Cost Effectiveness Ratios, and cost per quality-adjusted life-year, 12 weeks following randomisation.

Other outcomes of cost-per-case of any improvement in hearing and cost-per-case for the different increments of hearing improvement will also be reported. Given the skewness inherent in most cost data and the concern of economic analyses with mean costs, we shall use a bootstrapping approach in order to calculate confidence intervals around the difference in mean costs [20].

The incremental economic analysis will be conducted on both the HUI3 and ICECAP-A, with confidence intervals generated to estimate uncertainty. The results of these economic analyses will be presented using cost-effectiveness acceptability curves to reflect sampling variation and uncertainties in the appropriate threshold cost-effectiveness value. Both deterministic and stochastic cost effectiveness analyses will be used to explore the robustness of these results to plausible variations in key assumptions and variations in the analytical methods used, and to consider the broader issue of the generalisability of the results.

## Ethical considerations

The trial will be conducted in accordance with the UK Policy Framework for Health and Social Care Research 2017, the applicable UK Acts of Parliament and Statutory Instruments. The protocol has been approved by an NHS Health Research Authority Research Ethics Committee (IRAS number 1004878, approved on 07/11/2022). Any necessary protocol modifications will be first approved by the sponsor and ethics committee prior to cascading to sites. Before any participants are enrolled into the trial, the Principle Investigator (PI) at each site is required to obtain the necessary local approval.

Participants with confirmed ISSNHL will be approached about the trial by their clinical care team during the standard clinical appointment following referral to secondary care. A Participant Information Sheet (PIS) will be provided by the trial team (S3 File), and participants will also be invited to watch a short explanatory video on their own smartphone, tablet or hospital computer at https://entintegrate.co.uk/starfish. Electronic documentation of consent will then be obtained from all participants by a member of the trial team (S4 File). Opt-in additional consents will be sought for home hearing test participation and data sharing with other researchers. At each visit the participant's willingness to continue in the trial will be ascertained and participants will maintain the right to withdraw from the trial at any point. Given the urgent need to commence treatment for the condition, randomisation and commencement of treatment should ideally occur on the same or next working day after written consent has been gained.

## Public and patient involvement

Patients and public members have been involved from the start of the trial design, in particular championing the adoption of several outcome measures, including speech, tinnitus and dizziness questionnaires, and the optional participant-led online hearing testing. A patient representative is part of the Trial Management Group (TMG), and the STARFISH patient group has developed written materials for the trial, assessed online testing and will assist in developing a summary of the results.

## Status and timeline of the study

The trial is in the setup phase. It is anticipated that the first participants will be enrolled in July 2023, with an 18 month recruitment period, and additional three month period to complete follow up and 6 months for analysis and closure.

## Organisational structure

The Sponsor for this trial is the University of Birmingham and the coordinating centre is Birmingham Clinical Trials Unit. Participant recruitment and data collection will be coordinated by a responsible Principal Investigator (PI) at each site, assisted by an NIHR Associate PI, in most cases an ENT trainee, at most sites. INTEGRATE, the UK ENT Trainee Research Network, will coordinate associate PI involvement. The following groups have been established to manage and monitor the STARFISH trial:

**Trial Management Group.** The TMG comprises individuals responsible for the day-to-day management of the trial: the co-chief investigators, co-investigators, statisticians, trial team leader, trial manager and lay members. The role of the group is to monitor all aspects of the conduct and progress of the trial, ensure that the protocol is adhered to and take appropriate action to safeguard participants and the quality of the trial itself.

**Trial Steering Committee.** The Trial Steering Committee (TSC) comprises independent and non-independent members including those with a background in clinical practice, methodology, health economics and statistics. The TSC will provide oversight of the trial, monitoring trial progress and conduct, and advises the TMG where required.

**Data Monitoring Committee.** The role of the independent Data Monitoring Committee (DMC) is to monitor the trial data, and to make recommendations to the TSC on whether there are any ethical or safety reasons as to why the trial should not continue or whether it needs to be modified. Data on safety outcomes and primary and major secondary outcomes will be supplied to the DMC during the trial.

## Discussion

The STARFISH trial aims to determine which route of steroid administration should be adopted as first-line treatment for ISSNHL. Elements of the design are pragmatic, to enhance recruitment and ensure relevance to current practice, for example with permitted variation in the timing and delivery of the interventions. Other variables had to be fixed to ensure the trial is able to detect superiority of a route of administration. The rationale behind these choices is outlined below.

### Choice of participants

An established definition for ISSNHL exists, however there is uncertainty over the appropriate treatment window from a hearing loss occurring [1, 21]. The chosen four-week recruitment window from symptom onset reflects current practice, where more than half of UK surgeons will prescribe intratympanic steroids up to one month after onset [9]. Two meta-analyses also suggest the final hearing threshold may be independent of the delay in steroid treatment, for both primary [21] and secondary therapy [22].

### Choice of interventions

The three arms have been selected to reflect the variation in current practice, with all interventions already in widespread clinical use. Oral steroid (prednisolone) is included as an intervention as this is recommended in guidelines [16], and used by most ENT clinicians across the UK [23, 24]. As the current standard of care, this represents the control intervention.

Intratympanic dexamethasone is included as a recent survey suggested 62% of UK ENT surgeons use intratympanic steroid injection in the treatment of ISSNHL [24], with 80% of surgeons spacing injections 2–7 days apart [24]. Meta-analysis shows no evidence that one intratympanic steroid regime is superior [21], and so the most commonly used agent

(dexamethasone) and numbers of injections (three) in the UK have been adopted [24]. For surgeons not currently using intratympanic injection for ISSNHL, the technique is easy to learn, and training on the technique for the STARFISH trial will be provided via pre-recorded video materials or on-site if required.

The third intervention is combined oral and intratympanic steroids, which has been included due to limited evidence that combination therapy may provide superior hearing recovery to monotherapy [4].

## Choice of outcomes

The choice of primary outcome, the improvement of PTA, has been recommended in an ISSNHL consensus statement [3] and is used in most studies for ISSNHL. Meta-analysis suggests that the time of endpoint measurement has no effect on the change in PTA [21], but as hearing recovery is rarely seen after 12 weeks [25], the PTA at 12 weeks has been chosen as the primary endpoint.

Secondary outcomes are a range of functional measures of hearing, including speech discrimination scores and patient-reported measures. Online home hearing tests have been developed to allow testing at short intervals that would not be practical otherwise, and the tests are strongly supported by our patient collaborators. Participants will be supplied with headphones and test results will be validated by comparison to in-hospital audiograms. These tests should provide greater detail on the rate of hearing recovery in each treatment arm at different time points.

## Patient access to treatment

Patient and public involvement work by the trial team identified delays in patient presentation to healthcare, and incorrect initial diagnosis that delayed patient access to specialist care in some cases. To enhance patient presentation, the STARFISH team, working with a patient group, has developed advertising for social media that alerts patients to the need for testing and urgent treatment if symptoms suggest ISSNHL. In addition, a poster has been created to be displayed in community medical and audiology practices.

In the UK most individuals with ISSNHL initially present to their community General Practitioner (GP), where ISSNHL may not initially be identified. To aid the early diagnosis of ISSNHL, video and written educational materials aimed at GPs have been developed. The STARFISH team have also created a smartphone and website app to aid less experienced clinicians in performing and interpreting the Weber tuning fork test. This provides a screening test that can be used to distinguish between a conductive hearing loss and a sensorineural loss that may require urgent specialist assessment [1].

## Team structure and recruitment strategy

UK national guidance states that ISSNHL should be referred urgently to an on-call ENT team [2]. Referrals may occur outside working hours, and treatment is frequently initiated at the time of diagnosis by the on-call team. The STARFISH trial therefore cannot rely on a conventional research team structure and screening process. For this reason, it is strongly recommended that at each trial site the PI appoints an associate PI who is an ENT specialist trainee, who is better able to access patients and coordinate trainee colleagues and therefore improve recruitment. INTEGRATE, the UK ENT Trainee Research Network will use its established network to manage this trainee involvement. Associate PIs will receive training and formal recognition within the NIHR Associate PI scheme.

Another barrier to recruitment is general practitioners recognising that the hearing loss is sensorineural rather than conductive and making an urgent referral to ENT. We have produced an app to help GPs make the correct diagnosis as well as an educational video and have partnered with GP research groups. We have established a Facebook group to directly target patients with SSNHL to give them information and encourage them to seek urgent medical input.

## Limitations of the study

There is a significant spontaneous hearing recovery rate in patients with ISSNHL who are untreated [4, 21], and quantifying this recovery would be of particular value if no route of steroid delivery was found to be superior. However, we have not incorporated a placebo arm as it would be unethical to withhold standard care with steroids [3, 23], and during patient and public consultation, potential trial participants said they would not accept randomisation to no treatment, making a trial including a placebo arm non-viable. It would be possible to give placebo tablets to the intratympanic steroid only group, and a placebo/sham injection to the tablet-only group, but in the case of injection this would increase participant risk without potential therapeutic benefit. Incorporating placebo treatments would also result in an increase in study cost and complexity, with potentially limited gain. It was concluded that the single blinding approach was adequate, given the relatively objective nature of the primary outcome in the form of a pure tone audiogram. It is accepted however that certain outcomes, such as those collected using patient reported outcome measures, could be impacted by participant awareness of their allocated intervention.

Variables that may be associated with hearing recovery in ISSNHL are not well documented. STARFISH uses a minimisation algorithm at the point of randomisation to ensure that the number of participants with variables known to be related to the extent of hearing recovery are balanced between arms, but it is possible that other variables may impact recovery and not be controlled for. The variables incorporated in the minimisation algorithm are: hearing severity, as there is a correlation between the hearing threshold before treatment and recovery [21]; time to treatment, as progressive spontaneous recovery may mean those treated later have already undergone some recovery; and presence of vertigo, as this is likely an indicator of poor prognosis [26, 27]. Allocation to interventions is also matched for each centre, given potential differences between sites in delivery of the interventions and local patient population.

## Supporting information

**S1 Checklist. SPIRIT checklist.**
(PDF)

**S1 File. Outcome assessment.**
(PDF)

**S2 File. Adverse event reporting.**
(PDF)

**S3 File. Participant information sheet.**
(PDF)

**S4 File. Informed consent form.**
(PDF)

**S5 File. Oral steroid participant information sheet.**
(PDF)

**S6 File. Intratympanic steroid participant information sheet.**
(PDF)

**S1 Protocol. Full study protocol.**
(PDF)

## Acknowledgments

The online home hearing tests were developed in collaboration with the HearX Group, Pretoria, South Africa. We thank the following for their ongoing involvement in the STARFISH trial: Saima Rajasingam, John Hardman, Chloe Swords, Annika Feilbach, Neil Winkles, Serge Engamba and Mahmoud Keshavarzi.

We would like to thank our Data Monitoring Committee and Trial Steering Committee members and the principal investigators (PI) at our study sites. We are particularly grateful to our future study participants.

We would like to thank the NIHR HTA for funding the study. We are also most grateful for the support of the sponsor, University of Birmingham, and the University of Birmingham Clinical Trials Unit. We also acknowledge the support of the NIHR Cambridge Biomedical Research Centre.

The views expressed in this publication are those of the author(s) and not necessarily those of the NHS, the National Institute for Health Research, Health Education England or the Department of Health and Social Care.

## Author Contributions

**Conceptualization:** Matthew E. Smith, Manohar L. Bance, James R. Tysome.

**Data curation:** Samir Mehta, Yongzhong Sun, Karen James.

**Formal analysis:** Matthew E. Smith, Samir Mehta, Yongzhong Sun, Tracy E. Roberts, James R. Tysome.

**Funding acquisition:** Matthew E. Smith, Rachel Knappett, Deborah Vickers, David White, Chris J. Schramm, Samir Mehta, Manohar L. Bance, James R. Tysome.

**Investigation:** Matthew E. Smith, Rachel Knappett, Deborah Vickers, David White, Chris J. Schramm, Samir Mehta, Yongzhong Sun, Marie Chadburn, Hugh Jarrett, Karen James, Tracy E. Roberts, Manohar L. Bance, James R. Tysome.

**Methodology:** Matthew E. Smith, Rachel Knappett, Deborah Vickers, Chris J. Schramm, Samir Mehta, Yongzhong Sun, Ben Watkins, Marie Chadburn, Hugh Jarrett, Karen James, Elizabeth Brettell, Tracy E. Roberts, Manohar L. Bance, James R. Tysome.

**Project administration:** Hugh Jarrett, Karen James, Elizabeth Brettell.

**Resources:** Matthew E. Smith, David White, Karen James, James R. Tysome.

**Supervision:** Matthew E. Smith, James R. Tysome.

**Writing – original draft:** Matthew E. Smith, Samir Mehta, James R. Tysome.

**Writing – review & editing:** Matthew E. Smith, Rachel Knappett, Deborah Vickers, David White, Chris J. Schramm, Samir Mehta, Yongzhong Sun, Ben Watkins,

Marie Chadburn, Hugh Jarrett, Karen James, Elizabeth Brettell, Tracy E. Roberts, Manohar L. Bance, James R. Tysome.

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
