## [Decision Letter · Decision Letter 0]

28 Sep 2023

PONE-D-23-24534

Protocol for a multicentre randomised controlled trial of STeroid Administration Routes For Idiopathic Sudden sensorineural Hearing loss: the STARFISH trial

PLOS ONE

Dear Dr. Smith,

Thank you for submitting your manuscript to PLOS ONE. After careful consideration, we feel that it has merit but does not fully meet PLOS ONE’s publication criteria as it currently stands. Therefore, we invite you to submit a revised version of the manuscript that addresses the points raised during the review process.

 Minor methodological revision required

We look forward to receiving your revised manuscript.

Kind regards,

Jeyasakthy Saniasiaya, MD, MMed ORLHNS, FEBORLHNS

Academic Editor

PLOS ONE

Journal Requirements:

3. We note that the original protocol that you have uploaded as a Supporting Information file contains an institutional logo. As this logo is likely copyrighted, we ask that you please remove it from this file and upload an updated version upon resubmission.

Additional Editor Comments:

The study protocol is well written.

Please justify the choice of drugs and the chosen concentration

Provide explanation why tinnitus and vertigo were not assessed in this study

Comments from PLOS Editorial Office:

We note that one or more reviewers has recommended that you cite specific previously published works. As always, we recommend that you please review and evaluate the requested works to determine whether they are relevant and should be cited. It is not a requirement to cite these works. We appreciate your attention to this request.

Reviewers' comments:

Reviewer's Responses to Questions

**Comments to the Author**

1. Does the manuscript provide a valid rationale for the proposed study, with clearly identified and justified research questions?

Reviewer #1: Yes

Reviewer #2: Yes

2. Is the protocol technically sound and planned in a manner that will lead to a meaningful outcome and allow testing the stated hypotheses?

Reviewer #1: Yes

Reviewer #2: Yes

3. Is the methodology feasible and described in sufficient detail to allow the work to be replicable?

Reviewer #1: Yes

Reviewer #2: Yes

4. Have the authors described where all data underlying the findings will be made available when the study is complete?

Reviewer #1: Yes

Reviewer #2: Yes

5. Is the manuscript presented in an intelligible fashion and written in standard English?

Reviewer #1: Yes

Reviewer #2: Yes

6. Review Comments to the Author

You may also provide optional suggestions and comments to authors that they might find helpful in planning their study.

Reviewer #1: This is a proposal of a prospective, randomized, interventional, comparative non-inferiority study. The introduction, rationale and aims are valid. I have no concerns regading internal or external validity.

The introduction is fine, and regarding methodology, I would urge the authors to consider the followinf suggestions:

1. In the Randomisation algorithm, I would suggest adding a tinnitus category (present prior, new-onset, and reduction during treatment), as it is known that different treatment modalities affect tinnitus diferently.

2. The schedule of intratympanic injections should be outlined with regard to oral dosing, to be able to discern drug dose/effect ratio.

3. The PTA prior to IT administration is a fine addition meant to prevent discomfort to the patient, but introduces sampling and measurement bias. I would suggest that all patients receive three injections, to ensure comparability of data.

4. Limiting improvement of PTA to 4 kHz limits the reach of analysis as a primary measure. Traditionally, speech-discriminating frequencies are seen as the most important, but high-frequency hearing loss often causes disturbing tinnitus, and I would suggest including 0.5 - 6 and 8 kHz, as a secondary outcome, to account for that, and then again separately, as proposed in point 4 of functional hearing outcomes section.

The sample size, adherence and statistic testing are fine.

Regarding placebo trials and comparison, as well as outcome testing, I would suggest citing the following study: Ajduk J, Peček M, Kelava I, Žaja R, Ries M, Košec A. Comparison of Intratympanic Steroid and Hyperbaric Oxygen Salvage Therapy Hearing Outcomes in Idiopathic Sudden Sensorineural Hearing Loss: A Retrospective Study. Ear Hear. 2023 Jul-Aug 01;44(4):894-899. doi: 10.1097/AUD.0000000000001338. Epub 2023 Jan 20. PMID: 36693145.

Reviewer #2: The study protocol of the multicentre RCT is well written and concise. However, some issues should be considered before publication:

The manuscript structure and order of reported items should be in line with the “Reporting checklist for protocol of a clinical trial.”, based on the SPIRIT guidelines. It is not clear why the authors deviated from that guideline or checklist.

It is not clear why the primary outcome is the PTA (0.5-4kHz), while the inclusion was an ISSHL measured at 3 contiguous frequencies. Why could the primary outcome not be aligned to that measure?

The intervention is not clear. Please specify the actual drugs to be used. Dexamethasone-phosphate has completely different molecular properties to dexamethasone and has different pharmacokinetic properties entering and leaving perilymph (see Salt & Plontke 2020 “Steroid Nomenclature in Inner Ear Therapy”. The same hold for the specification of prednisolone (Prenisolone 21-hemisuccinate, or Prednisolone-succinate?)

Please clarify the use of 3.3 or 3.8 mg/ml. The dose are quite similar but not equal. Please refer to Chandrasekhar et al 2019. Maybe 10 or better 24 mg/ml would be more appropriated.

Inclusion criteria: The 30-dB threshold is quite low and low effects of the interventions would be expected (see Fig. 3 in Chen, Halpin, Rauch 2003). It is not clear if the 30 dB are absolute thresholds or relative decrements. In this case please specify the reference. Is it an average over 3 (contig.) frequencies or at every single frequency? Table 1 says “0.5-8.0 kHz”…

The measurement of the VRBQ, TFI, HUI3 and ICECAP-A are very good choices to make the study results worth.

The „extent of hearing recovery“ has to be explained in more detail. Why was the 50% AB-phonems criterion used? At which sound pressure level?

Adverse events (Figure 1) have to be reported over the entire trial period, starting with the first drug application.

The study is not blinded and no placebo is used. This is a large risk for bias. Please discuss this issue also in the context of other similar trials, e.g., Plontke et al. 2022 ( https://doi.org/10.1007/s00106-022-01184-8 ).

7. PLOS authors have the option to publish the peer review history of their article (what does this mean?). If published, this will include your full peer review and any attached files.

Reviewer #1: No

Reviewer #2: No

---

## [Author Response · Author response to Decision Letter 0]

6 Oct 2023

PONE-D-23-24534: Response to peer review

Protocol for a multicentre randomised controlled trial of STeroid Administration Routes For Idiopathic Sudden sensorineural Hearing loss: the STARFISH trial

We thank the academic editor and reviewers for their comments. We have addressed these individually below.

Additional Editor Comments:

The study protocol is well written.

Please justify the choice of drugs and the chosen concentration

Dexamethasone is the most commonly used intratympanic steroid in the UK (Lechner M, Sutton L, Ferguson M, Abbas Y, Sandhu J, Shaida A. Intratympanic Steroid Use for Sudden Sensorineural Hearing Loss: Current Otolaryngology Practice. The Annals of otology, rhinology, and laryngology. 2019;128(6):490-502), and is widely used both clinically and in existing studies (Plontke SK, Meisner C, Agrawal S, Cayé-Thomasen P, Galbraith K, Mikulec AA, Parnes L, Premakumar Y, Reiber J, Schilder AG, Liebau A. Intratympanic corticosteroids for sudden sensorineural hearing loss. Cochrane Database Syst Rev. 2022 Jul 22;7(7):CD008080. doi: 10.1002/14651858.CD008080.pub2. PMID: 35867413; PMCID: PMC9307133.) as well as having good availability globally. Meta-analysis and simulation shows no evidence that one intratympanic steroid agent or dosing regime is superior over another (Liebau A, Pogorzelski O, Salt AN, Plontke SK. Hearing Changes After Intratympanically Applied Steroids for Primary Therapy of Sudden Hearing Loss: A Metaanalysis Using Mathematical Simulations of Drug Delivery Protocols. Otol Neurotol. 2017;38(1):19-30.).

The team therefore chose dexamethasone at the dose of 3.3mg/ml or 3.8mg/ml as it is widely available and clinically used, with an established safety profile and basis for use in the literature. The results of the study will therefore be directly clinically applicable, and any recommendations or resulting policy rapidly translated into practice. 

Regarding the dose choice for dexamethasone, the decision to allow two concentrations of the same drug is a pragmatic one. In the UK most hospitals only stock 3.3mg/ml (the 3.3mg refers to the base rather than the salt). This formulation is labelled 4mg dexamethasone phosphate. Most manufacturers sell the 3.3mg/ml but there are a couple who produce the 3.8mg/ml base (5mg dexamethasone phosphate). 

The dose delivered is volume-based depending on the size of the middle ear cleft, and this may vary considerably between individuals. In addition, the ability to fill the middle ear will depend on location of tympanic membrane puncture and patient positioning, and some patients may clear the drug more quickly depending on Eustachian tube function once sitting up. There is a training video to standardise steroid injection, but there will inevitably be some variation. Given the small volume injected and multiple other variables, allowing the use of both 3.3mg/ml or 3.8mg/ml dexamethasone was thought to be reasonable to facilitate the running of the trial and was felt unlikely to be the biggest determinant of dose given. 

The rationale is summarised in Discussion/Choice of interventions. 

Provide explanation why tinnitus and vertigo were not assessed in this study

Vertigo forms one of the minimisation criteria for matching trial arms.

Tinnitus and dizziness are key secondary outcomes. They are assessed at baseline and all follow up points via the Tinnitus Functional Index (TFI) and Vestibular Rehabilitation Benefit Questionnaire (VRBQ). 

In addition a measure of tinnitus and dizziness severity will be recorded via visual analogue score (slider) on a weekly basis for those participants using the home hearing test software.

Comments from PLOS Editorial Office:

We note that one or more reviewers has recommended that you cite specific previously published works. As always, we recommend that you please review and evaluate the requested works to determine whether they are relevant and should be cited. It is not a requirement to cite these works. We appreciate your attention to this request.

Reviewers' comments:

Reviewer's Responses to Questions

Comments to the Author

1. Does the manuscript provide a valid rationale for the proposed study, with clearly identified and justified research questions?

Reviewer #1: Yes

Reviewer #2: Yes

2. Is the protocol technically sound and planned in a manner that will lead to a meaningful outcome and allow testing the stated hypotheses?

Reviewer #1: Yes

Reviewer #2: Yes

3. Is the methodology feasible and described in sufficient detail to allow the work to be replicable?

Reviewer #1: Yes

Reviewer #2: Yes

4. Have the authors described where all data underlying the findings will be made available when the study is complete?

Reviewer #1: Yes

Reviewer #2: Yes

5. Is the manuscript presented in an intelligible fashion and written in standard English?

Reviewer #1: Yes

Reviewer #2: Yes

6. Review Comments to the Author

You may also provide optional suggestions and comments to authors that they might find helpful in planning their study.

Reviewer #1: This is a proposal of a prospective, randomized, interventional, comparative non-inferiority study. The introduction, rationale and aims are valid. I have no concerns regarding internal or external validity.

The introduction is fine, and regarding methodology, I would urge the authors to consider the followinf suggestions:

1. In the Randomisation algorithm, I would suggest adding a tinnitus category (present prior, new-onset, and reduction during treatment), as it is known that different treatment modalities affect tinnitus diferently.

We have limited minimisation criteria to variables known or suspected to impact the primary outcome, which is pure tone audiometry. To our knowledge, the presence of tinnitus is not identified as associated with poor or enhanced hearing recovery, beyond an association with severity of hearing loss, which does form one of the minimisation variables.

2. The schedule of intratympanic injections should be outlined with regard to oral dosing, to be able to discern drug dose/effect ratio.

As outlined in Methods/Interventions, the first intratympanic injection will be given within ±4 days of starting oral steroids. This flexibility has been built in following consultation with trial sites, as in some sites practical considerations relating to staff and facilities mean further defining the interval would lead to more protocol deviations. Based on feedback we do however expect most participants in the combined intervention arm to receive both oral and intratympanic injections on the same day. Figure 1 (SPIRIT schedule) also provides detail on intervention timing. 

Timing of dosing will be captured in the electronic data capture forms. We will report actual achieved dosing schedules in the primary trial publication. 

3. The PTA prior to IT administration is a fine addition meant to prevent discomfort to the patient, but introduces sampling and measurement bias. I would suggest that all patients receive three injections, to ensure comparability of data.

This is a valid consideration that we have given thought to. We do not intend to analyse the pre-injection PTA data other than in terms of adherence. It is unclear why there may be measurement bias. We do accept that there may be some site-to-site variability. 

The current protocol reflects current clinical practice, and 3 injections will be delivered unless hearing has fully recovered. It is pragmatic, intentionally replicating standard care, for ease of delivery, acceptability to patients and clinicians and translatability to practice. We understand that in the event of full hearing recovery clinicians and patients would be hesitant to undertake the risk of further injection, and we seek therefore to include as many patients as possible with the current design. 

4. Limiting improvement of PTA to 4 kHz limits the reach of analysis as a primary measure. Traditionally, speech-discriminating frequencies are seen as the most important, but high-frequency hearing loss often causes disturbing tinnitus, and I would suggest including 0.5 - 6 and 8 kHz, as a secondary outcome, to account for that, and then again separately, as proposed in point 4 of functional hearing outcomes section.

As stated by the reviewer, high frequency hearing thresholds measured by the absolute improvement in pure tone audiogram average across 4.0, 6.0 and 8.0 kHz will be reported as a secondary outcome, given their importance in functional hearing. The frequency range for the primary outcome measure chosen reflects that used in other trials on sudden sensorineural hearing loss, enabling future meta-analysis. 

The sample size, adherence and statistic testing are fine.

Regarding placebo trials and comparison, as well as outcome testing, I would suggest citing the following study: Ajduk J, Peček M, Kelava I, Žaja R, Ries M, Košec A. Comparison of Intratympanic Steroid and Hyperbaric Oxygen Salvage Therapy Hearing Outcomes in Idiopathic Sudden Sensorineural Hearing Loss: A Retrospective Study. Ear Hear. 2023 Jul-Aug 01;44(4):894-899. doi: 10.1097/AUD.0000000000001338. Epub 2023 Jan 20. PMID: 36693145.

We were guided in our study design by the international consensus paper for studies on ISSNHL (Marx M, Younes E, Chandrasekhar SS, Ito J, Plontke S, O'Leary S, et al. International consensus (ICON) on treatment of sudden sensorineural hearing loss. Eur Ann Otorhinolaryngol Head Neck Dis. 2018;135(1S):S23-S8). Thank you for highlighting the interesting study you cite, which identifies the main speech-discriminating frequencies as 500 Hz to 4000 Hz, as are covered by our primary outcome. 

Reviewer #2: The study protocol of the multicentre RCT is well written and concise. However, some issues should be considered before publication:

The manuscript structure and order of reported items should be in line with the “Reporting checklist for protocol of a clinical trial.”, based on the SPIRIT guidelines. It is not clear why the authors deviated from that guideline or checklist.

The order is that recommended by PLOS one. The SPIRIT checklist is supplied completed with the initial submission. 

It is not clear why the primary outcome is the PTA (0.5-4kHz), while the inclusion was an ISSHL measured at 3 contiguous frequencies. Why could the primary outcome not be aligned to that measure?

The inclusion criteria and outcome are already aligned, both relating to frequencies of 0.5, 1.0, 2.0, 4.0kHz. The accepted international definition of sudden sensorineural hearing loss used in our study only requires 3 frequencies to be affected. The current choices align with other work and therefore permit meta-analysis. 

The intervention is not clear. Please specify the actual drugs to be used. Dexamethasone-phosphate has completely different molecular properties to dexamethasone and has different pharmacokinetic properties entering and leaving perilymph (see Salt & Plontke 2020 “Steroid Nomenclature in Inner Ear Therapy”. The same hold for the specification of prednisolone (Prenisolone 21-hemisuccinate, or Prednisolone-succinate?)

Thank you for this comment, we have amended the text to specify dexamethasone phosphate. In the UK this is the only form of dexamethasone available in the specified preparations. 

We consulted our trials pharmacy team regarding prednisolone. For standard tablets both we and they were unable to see any reference to the salt used in prescribing guidance, or in the British National Formulary. On review of stocked tablets and their packaging, no reference is made to the salt. We do note that the soluble tablets are prednisolone as prednisolone sodium phosphate, and as is oral solution. We therefore have not specified the salt. 

Please clarify the use of 3.3 or 3.8 mg/ml. The dose are quite similar but not equal. Please refer to Chandrasekhar et al 2019. Maybe 10 or better 24 mg/ml would be more appropriated.

Meta-analysis and simulation shows no evidence that one intratympanic steroid agent or dosing regime is superior over another (Liebau A, Pogorzelski O, Salt AN, Plontke SK. Hearing Changes After Intratympanically Applied Steroids for Primary Therapy of Sudden Hearing Loss: A Metaanalysis Using Mathematical Simulations of Drug Delivery Protocols. Otol Neurotol. 2017;38(1):19-30.). The team therefore chose dexamethasone at the dose of 3.3mg/ml or 3.8mg/ml as it is widely available and clinically used in the UK, with an established safety profile and basis for use in the literature. 

A pragmatic approach reflecting clinical practice, allowing either 3.3mg/ml or 3.8mg/ml, aids study delivery, without significantly compromising analysis. The results of the study will therefore be directly clinically applicable, and any recommendations or resulting policy rapidly translated into practice. 

The rationale is summarised in Discussion/Choice of interventions. 

Inclusion criteria: The 30-dB threshold is quite low and low effects of the interventions would be expected (see Fig. 3 in Chen, Halpin, Rauch 2003). It is not clear if the 30 dB are absolute thresholds or relative decrements. In this case please specify the reference. Is it an average over 3 (contig.) frequencies or at every single frequency? Table 1 says “0.5-8.0 kHz”. 

The most frequently used audiometric criterion for SSNHL is a decrease in hearing of 30 decibels affecting at least 3 consecutive frequencies. Because premorbid audiometry is generally unavailable, hearing loss is often defined in relation to the opposite ear’s thresholds (Chandrasekhar SS, Tsai Do BS, Schwartz SR, Bontempo LJ, Faucett EA, Finestone SA, et al. Clinical Practice Guideline: Sudden Hearing Loss (Update). Otolaryngol Head Neck Surg. 2019;161(1_suppl):S1-S45.)

For clarity, the inclusion criteria have been reworded: 

“Diagnosis of new-onset ISSNHL: a new increase in sensorineural thresholds of 30 decibels (dBHL) or greater with onset over a period of 3 days or less according to the patient's history, and affecting each of 3 contiguous pure-tone frequencies (out of 0.5, 1.0, 2.0, 4.0 kilohertz (kHz)) confirmed with a pure tone audiogram. Where audiometry is not available prior to the ISSNHL and there is a history is of equal hearing in both ears prior to the sudden loss, hearing loss will be defined in relation to the opposite ear’s thresholds. Where audiometry is not available prior to the ISSNHL and there is a history of different hearing in both ears prior to the sudden loss, then the candidate can only be included if the ISSNHL occurred in the better hearing ear and the measured thresholds are at least 30dB below the contralateral ear at 3 contiguous pure-tone frequencies (out of 0.5, 1.0, 2.0, 4.0 kilohertz (kHz)) confirmed with a pure tone audiogram.”

Apologies but we have been unable to find Table 1 stating “0.5-8.0 kHz”. A secondary outcome will be high frequency hearing measured at 4, 6 and 8kHz. 

The measurement of the VRBQ, TFI, HUI3 and ICECAP-A are very good choices to make the study results worth.

The „extent of hearing recovery“ has to be explained in more detail. Why was the 50% AB-phonems criterion used? At which sound pressure level?

The outcome requires 50% at any intensity. The assessment will start at 30dB above the PTA average and intensity will be increased until the maximum score is reached. This is detailed in the associated supplementary material full protocol, but has now also been added to the text in Materials and Methods/Outcomes. 

50% was chosen as this is generally agreed to be a predictor of hearing aid benefit, and has been used as such in AAO-HNS guidance and some SSNHL studies. The literature regarding SSNHL is limited in this area, but 50% is more commonly cited as being the minimum score to achieve success with hearing aids in vestibular schwannoma populations.

Adverse events (Figure 1) have to be reported over the entire trial period, starting with the first drug application.

The study is not blinded and no placebo is used. This is a large risk for bias. Please discuss this issue also in the context of other similar trials, e.g., Plontke et al. 2022 ( https://doi.org/10.1007/s00106-022-01184-8 ).

The study is single blinded and primary outcome measurements will be made by blinded assessors. Participant blinding is not feasible without placebo. 

The lack of placebo is discussed in Discussion/Limitations of the study. As outlined here, ‘during patient and public consultation, potential trial participants said they would not accept randomisation to no treatment, making a trial including a placebo-only arm non-viable’. 

We have expanded this section to include: “It would be possible to give placebo tablets to the intratympanic steroid only group, and a placebo/sham injection to the tablet-only group, but in the case of injection this would increase participant risk without potential therapeutic benefit. Incorporating placebo treatments would also result in an increase in study cost and complexity, with potentially limited gain. It was concluded that the single blinding approach was adequate, given the relatively objective nature of the primary outcome in the form of a pure tone audiogram. It is accepted however that certain outcomes, such as those collected using patient reported outcome measures, could be impacted by participant awareness of their allocated intervention.” 

Thank you for highlighting the cited study. There are no doubt cultural and other differences that may permit this work in other countries, and participant acceptance is a fascinating aspect of study design. 

7. PLOS authors have the option to publish the peer review history of their article (what does this mean?). If published, this will include your full peer review and any attached files.

Do you want your identity to be public for this peer review? For information about this choice, including consent withdrawal, please see our Privacy Policy.

Reviewer #1: No

Reviewer #2: No

---

## [Decision Letter · Decision Letter 1]

15 Nov 2023

PONE-D-23-24534R1Protocol for a multicentre randomised controlled trial of STeroid Administration Routes For Idiopathic Sudden sensorineural Hearing loss: the STARFISH trialPLOS ONE

Dear Dr. Smith,

Thank you for submitting your manuscript to PLOS ONE. After careful consideration, we feel that it has merit but does not fully meet PLOS ONE’s publication criteria as it currently stands. Therefore, we invite you to submit a revised version of the manuscript that addresses the points raised during the review process.

ACADEMIC EDITOR: Minor revision as suggested below

We look forward to receiving your revised manuscript.

Kind regards,

Jeyasakthy Saniasiaya, MD, MMed ORLHNS, FEBORLHNS

Academic Editor

PLOS ONE

Journal Requirements:

Additional Editor Comments:

Well-written paper and revised adequately. Minor suggestions for improvement as listed below

Reviewers' comments:

Reviewer's Responses to Questions

**Comments to the Author**

1. Does the manuscript provide a valid rationale for the proposed study, with clearly identified and justified research questions?

Reviewer #1: Yes

Reviewer #2: Yes

Reviewer #3: Yes

2. Is the protocol technically sound and planned in a manner that will lead to a meaningful outcome and allow testing the stated hypotheses?

Reviewer #1: Yes

Reviewer #2: Yes

Reviewer #3: Yes

3. Is the methodology feasible and described in sufficient detail to allow the work to be replicable?

Reviewer #1: Yes

Reviewer #2: Yes

Reviewer #3: Yes

4. Have the authors described where all data underlying the findings will be made available when the study is complete?

Reviewer #1: Yes

Reviewer #2: Yes

Reviewer #3: No

5. Is the manuscript presented in an intelligible fashion and written in standard English?

Reviewer #1: Yes

Reviewer #2: Yes

Reviewer #3: Yes

6. Review Comments to the Author

You may also provide optional suggestions and comments to authors that they might find helpful in planning their study.

Reviewer #1: Thank you for the revision, I have no further issues to discuss. The authors have made all of the required changes and I would be happy to support publication.

Reviewer #2: Dear authors. Thank you for considering the reviewer comments. I fully understand that substantial modifications of the study protocol are not possible anymore. My question about my role as reviewer in this context was unfortunalty not answered by the editors. If focussing on the manuscript I have no further comments and wish good success when conducting the study!

Reviewer #3: In this study protocol, a three-arm randomized control trial with an initial pilot study is being proposed. The primary outcome will be absolute improvement in pure tone audiogram at 12-weeks. Cost effectiveness will also be compared among the three arms.

Minor revisions:

1- The standard statistical term for average is mean.

2- Indicate if adverse events will be collected according to a standardized format.

3- In the mixed-effect linear model indicate the type of underlying covariance structure that will be used or the method for selecting it.

4- For the adjusted Cox proportional hazards models, state the variables that will be used for adjusting or the method that will be applied to determine which variable will be included.

5- Identify the software that will be used to capture the data as well as the software that will be used for the statistical analysis.

6- To assist in the review process, add line numbers to the document.

7. PLOS authors have the option to publish the peer review history of their article (what does this mean?). If published, this will include your full peer review and any attached files.

Reviewer #1: **Yes: **Andro Košec, MD, PhD, FEBORL-HNS

Reviewer #2: No

Reviewer #3: No

---

## [Author Response · Author response to Decision Letter 1]

8 Dec 2023

PONE-D-23-24534: Response to peer review

Protocol for a multicentre randomised controlled trial of STeroid Administration Routes For Idiopathic Sudden sensorineural Hearing loss: the STARFISH trial

We thank the academic editor and reviewers for their further review of our manuscript. We have addressed these individually below.

6. Review Comments to the Author

Reviewer #1: Thank you for the revision, I have no further issues to discuss. The authors have made all of the required changes and I would be happy to support publication.

Reviewer #2: Dear authors. Thank you for considering the reviewer comments. I fully understand that substantial modifications of the study protocol are not possible anymore. My question about my role as reviewer in this context was unfortunatly not answered by the editors. If focussing on the manuscript I have no further comments and wish good success when conducting the study!

Reviewer #3: In this study protocol, a three-arm randomized control trial with an initial pilot study is being proposed. The primary outcome will be absolute improvement in pure tone audiogram at 12-weeks. Cost effectiveness will also be compared among the three arms.

Minor revisions:

1- The standard statistical term for average is mean.

While we agree, pure tone average is the commonplace term in both research and clinical practice when describing the outcome of audiometry and so has been used in this work. 

2- Indicate if adverse events will be collected according to a standardized format.

SAEs will be collected according to a standardised format on our SAE form and that AEs relevant to the interventions will be recorded according to a standardized format on our CRFs. 

3- In the mixed-effect linear model indicate the type of underlying covariance structure that will be used or the method for selecting it.

We will be using a compound symmetry covariance structure for the mixed-effect linear model. This has been added to the revised manuscript.

4- For the adjusted Cox proportional hazards models, state the variables that will be used for adjusting or the method that will be applied to determine which variable will be included.

All analysis will be adjusted for the minimisation variables as fixed effects (except site which will be included as a random effect). This has been added to the revised manuscript.

5- Identify the software that will be used to capture the data as well as the software that will be used for the statistical analysis.

Statistical analysis will be undertaken in Stata version 18 (or higher) and/or SAS software, version 9.4 (or higher). This has been added to the revised manuscript. 

6- To assist in the review process, add line numbers to the document.

Line numbers have been added

---

## [Decision Letter · Decision Letter 2]

12 Dec 2023

Protocol for a multicentre randomised controlled trial of STeroid Administration Routes For Idiopathic Sudden sensorineural Hearing loss: the STARFISH trial

PONE-D-23-24534R2

Dear Dr. Smith,

We’re pleased to inform you that your manuscript has been judged scientifically suitable for publication and will be formally accepted for publication once it meets all outstanding technical requirements.

Kind regards,

Jeyasakthy Saniasiaya, MD, MMed ORLHNS, FEBORLHNS

Academic Editor

PLOS ONE

Additional Editor Comments (optional):

Authors have adequately revised

Reviewers' comments:

Reviewer's Responses to Questions

**Comments to the Author**

1. Does the manuscript provide a valid rationale for the proposed study, with clearly identified and justified research questions?

Reviewer #1: Yes

Reviewer #2: Yes

Reviewer #3: Yes

2. Is the protocol technically sound and planned in a manner that will lead to a meaningful outcome and allow testing the stated hypotheses?

Reviewer #1: Yes

Reviewer #2: Yes

Reviewer #3: Yes

3. Is the methodology feasible and described in sufficient detail to allow the work to be replicable?

Reviewer #1: Yes

Reviewer #2: Yes

Reviewer #3: Yes

4. Have the authors described where all data underlying the findings will be made available when the study is complete?

Reviewer #1: Yes

Reviewer #2: Yes

Reviewer #3: No

5. Is the manuscript presented in an intelligible fashion and written in standard English?

Reviewer #1: Yes

Reviewer #2: Yes

Reviewer #3: Yes

6. Review Comments to the Author

You may also provide optional suggestions and comments to authors that they might find helpful in planning their study.

Reviewer #1: The revised version of the manuscript is accepteable for publication in its current form and I have no further issues to discuss with the authors.

Reviewer #2: This is a revised version of a previously reviewed manuscript with minor changes. I have no further comments.

Reviewer #3: All comments have been adequately addressed.

7. PLOS authors have the option to publish the peer review history of their article (what does this mean?). If published, this will include your full peer review and any attached files.

Reviewer #1: **Yes: **Andro Košec, MD, PhD, FEBORL-HNS

Reviewer #2: No

Reviewer #3: No

---

## [Editor Report · Acceptance letter]

14 Feb 2024

PONE-D-23-24534R2 

PLOS ONE

Dear Dr. Smith, 

I'm pleased to inform you that your manuscript has been deemed suitable for publication in PLOS ONE. Congratulations! Your manuscript is now being handed over to our production team.

Kind regards, 

on behalf of

Dr. Jeyasakthy Saniasiaya 

Academic Editor

PLOS ONE